# SpikeLoRA: Learnable Activation Sparsity for Low-Rank Adaptation using Spiking Neural Networks

## Abstract

Low-rank adaptation (LoRA) is a fine-tuning method that freezes the parameters of a pre-trained model and injects small trainable matrices. LoRA-based methods focus on parameter-level efficiency, but do not directly control the activations in the low-rank space. We introduce SpikeLoRA, a spiking low-rank adaptation fine-tuning method that leverages the leaky integrate-and-fire (LIF) neuron to introduce learnable sparsity with minimal computational overhead. The LIF neuron gates the activations from the $A$-matrix in LoRA, sparsifying them while preserving learned information. This design makes SpikeLoRA a sparse fine-tuning method for both spiking and traditional LLMs, with the additional efficiency benefit of being compatible with neuromorphic hardware. Our experiments show that over 70% sparsity is achievable without a significant drop in performance. Further, improved performance as compared to LoRA is observed for smaller datasets and higher-rank settings. We also show that SpikeLoRA indirectly mitigates overfitting, particularly for higher ranks.

## 1 Introduction

Fine-tuning forms part of the transfer learning domain (Raffel et al., 2020), and allows a pre-trained model to specialise in a downstream task, incorporating a specific domain of expertise, task, or knowledge. To achieve this, fine-tuning adjusts the weights of a pre-trained model to minimise some loss on a downstream task.

The problem with fine-tuning, however, is that parameters of the original model have to be updated (retrained). This results in computational inefficiencies and potential catastrophic forgetting, where pre-trained knowledge may be lost (Song et al., 2025). Adapter modules, or adapters, were proposed as a solution to fine-tuning inefficiencies, and add small fully-connected networks on top of the frozen pre-trained parameters (Houlsby et al., 2019). However, each downstream task requires its own adapter, making it difficult to switch tasks easily. Low-rank Adaptation (LoRA) is a novel fine-tuning approach which freezes the weights of the pre-trained model and uses low-rank matrix decomposition to parameterise the weight update (Hu et al., 2022). This substantially reduces the trainable parameters, does not introduce additional inference, and eliminates the need to calculate gradients for frozen parameters.

LoRA has evolved to multiple methods, including adaptive low-rank adaptation (AdaLoRA) (Zhang et al., 2023), adaptive learning low-rank adaptation (ALLoRA) (Huang & Balestriero, 2024), weight-decomposed low-rank adaptation (DoRA) (Liu et al., 2024), and quantised low-rank adaptation (QLoRA) (Dettmers et al., 2023), among others. These methods attempt to enhance LoRA's efficiency without compromising performance by introducing adaptive low-rank updates, weight decomposition, and quantisation techniques. Fundamentally, AdaLoRA dynamically adjust the rank of each respective LoRA module, ALLoRA eliminates dropout and scaling by introducing an adaptive learning rate, DoRA stabilises training by controlling weight redundancy, and QLoRA quantises the base model's weights while fine-tuning with LoRA. Furthermore, Huang & Balestriero (2024) investigated an adaptive scaling factor for LoRA (ASF-LoRA). Different from the constant scaling factor in Hu et al. (2022), ASF-LoRA makes the scaling factor learnable, but introduces potential ripple effects across blocks, which degrade performance.

All of the above LoRA variants operate directly on the model's low-rank weight matrices. In this work, we argue that efficiency is achievable not only on *parameter* level, but also on the *activation* level. Dropout (Srivastava et al., 2014) is a well-known regularisation method that stochastically zeroes activations during training to prevent co-adaptation of neurons. Notably, dropout has been successfully implemented in the context of LoRA, and shown to reduce the generalisation gap (Lin et al., 2024). While dropout suppresses activations *stochastically*, a *learnable* activation suppression pattern may yield more targeted and effective sparsity, and yield further benefits for both efficiency and generalisation. Spiking neural networks (SNNs) offer a mechanic ideally suited for this task.

SNNs are designed to be more biologically plausible than traditional artificial neural networks (ANNs), and make use of event-driven (discrete) spikes to transmit information (Singh et al., 2020). This makes SNNs inherently sparse and eliminates the need for continuous activations. SNNs have been successfully utilised in spiking large language models (LLMs), such as SpikeGPT (Zhu et al., 2024). SNNs offer promising biologically-inspired capabilities, and are more energy-efficient when deployed on specialised neuromorphic hardware. Since SNNs are sparse in nature, they possess the mechanic to learn when to sparsify activations. We apply SNN mechanics to LoRA and show that learnable sparsity can act as a compression (i.e., sparsification) mechanism with potential benefits of mitigating overfitting, especially for higher LoRA ranks.

In this paper, we contribute the following:

1. **SpikeLoRA:** A novel spiking version of LoRA is proposed, enabling more efficient fine-tuning on downstream tasks. By coupling the LoRA module with a leaky integrate-and-fire (LIF) neuron, biologically inspired parametric sparsification is introduced with minimal computational overhead. This allows SpikeLoRA to learn when to suppress activations while achieving accuracy comparable with classic LoRA.

2. **Application of LoRA and SpikeLoRA to SpikeGPT:** We fine-tune SpikeGPT (Zhu et al., 2024) using both LoRA and SpikeLoRA to show the possibility and potential of LoRA in a fully-spiking pipeline. Coupled with SpikeLoRA, we make the entire fine-tuning process compatible with neuromorphic hardware.

The rest of the paper is structured as follows: **Section 2** covers relevant background, including SNNs, SpikeGPT, and LoRA. **Section 3** formally introduces SpikeLoRA. **Section 4** presents and discusses the experimental setup and results. **Section 5** concludes the paper and outlines future work.

## 2 BACKGROUND

### 2.1 SPIKING NEURAL NETWORKS (SNNs)

SNNs are often referred to as the 3rd generation of neural networks (Capatina et al., 2023; Maass, 1997; Yang et al., 2024), and attempt to closely mimic the biological brain to solve known problems in deep learning, such as excessive memory usage, computational complexity (Eshraghian et al., 2023), and the lack of sufficient parallelism (Pfeiffer & Pfeil, 2018). Memory usage is reduced via the inherent sparsity of the SNNs. Computational complexity is reduced by using bio-inspired discrete spikes instead of continuous activations seen in ANNs. Parallelism is improved through the event-driven nature of SNNs.

In SNNs, neuron activation is driven by temporal binary signals, referred to as *spike trains*. An integrate-and-fire (IF) neuron ingests a spike train and accumulates the binary signals in the current membrane potential. When the membrane potential reaches a pre-defined voltage threshold, the neuron fires (i.e., activates), and the membrane potential is reset. The membrane potential will continue to build up until the voltage threshold is reached. If a neuron fires, it will contribute to the next neuron's membrane potential; otherwise, it acts as a silent neuron, which accounts for the sparsity of SNNs. Due to the dependency of membrane potential on past spikes, SNNs inherently possess recurrent properties. Neuromorphic hardware allows for true sparsity and event-driven activations, such that silent neurons do not use any memory, therefore offering a significant reduction in energy use.

### 2.1.1 Leaky Integrate-and-Fire (LIF) Neuron

Biological neurons lose their membrane potential over time, whereas IF neurons are incapable of doing so. To better model biological neurons, LIF neurons introduce a leaky aspect to capture the temporal effects. The first-order LIF neuron model has been widely used to understand and model SNNs (Kim et al., 2023). A first-order LIF neuron model (Dayan & Abbott, 2001) performs activation for time step $t$ by calculating membrane potential and comparing it to a set voltage threshold, $V_\theta$. The function $V_p[t]$ updates the membrane potential of a neuron per time step $t$ as follows:

$$V_p[t] = \beta \cdot V_p[t-1] + W \cdot X[t] - S[t-1] \cdot V_\theta, \tag{1}$$

where $\beta$ is a predefined decay factor (such as $e^{-1/\tau}$ (Eshraghian et al., 2023)) of $V_p[t-1]$, and $V_p[t-1]$ is the previous state of the neuron's membrane potential. $\beta$ is used to simulate the leaky aspect of an IF neuron. If $\beta = 1$, then Eq.(1) simply models a non-leaky IF neuron. $W \cdot X[t]$ is the weighted input to the LIF-neuron. $S[t-1]$ determines whether to reset the membrane potential, and is defined as follows:

$$S[t] = \Theta(V_p[t] - V_\theta), \tag{2}$$

where $\Theta$ is the Heaviside function (Legua et al., 2006). $S[t] \in \{0, 1\}$ since $\Theta \in \{0, 1\}$. Therefore, $V_p[t]$ (Eq.(1)) causes the membrane potential to accumulate when the neuron does not fire (i.e., where $V_p[t] < V_\theta$) (Tavanaei et al., 2019). When a neuron fires, *reset-by-subtraction* (Eq.(2)) will subtract the threshold, whereas *reset-to-zero* will reset the membrane potential to zero (Eshraghian et al., 2023).

For backpropagation, an arctangent surrogate gradient function has proven to be effective in approximating gradients (Eshraghian et al., 2023). The arctangent surrogate gradient solves the non-differentiable nature of the Heaviside function. However, since these functions approximate gradients, a loss in performance can be expected (Pfeiffer & Pfeil, 2018).

## 2.2 SpikeGPT

SpikeGPT (Zhu et al., 2024) is a generative spike-based LLM based on the receptance weighted key value (RWKV) architecture (Peng et al., 2023). SpikeGPT is suitable for natural language understanding (NLU) and natural language generation (NLG) tasks.

Rather than introducing an additional temporal dimension, Eshraghian et al. (2023) suggests directly adapting the neurons with spiking thresholds in the attention head to learn long-term dependencies. SpikeGPT's novel spiking RWKV (SRWKV) (Zhu et al., 2024) follows this approach by adapting neurons with spiking thresholds at the embedding layer in the RWKV architecture. SRWKV employs the same foundation as RWKV's time-mixing block, but to create a spiking version, it uses the recurrent properties of SNNs. SWRKV unrolls the sequence $X \in \mathbb{R}^{T \times d}$ to represent $X[t] \in \mathbb{R}^{1 \times d}$. Similar to RWKV, SRWKV uses $R$, $K$, and $V$ with linear transformations. These transformations are then used as inputs to the rest of the time-mixer block.

To adapt the feedforward network (FFN) block to be an SNN, Zhu et al. (2024) propose a spiking receptance FFN (SRFFN). The SRFFN functions similarly to the channel mixer. This is coupled with a spiking gating mechanism. SRFFN contains learnable parameters and utilises LIF neurons as resulting outputs.

To build a spike train, SpikeGPT uses binary embeddings (BEs) to transform continuous outputs into binary spikes (Zhu et al., 2024). The BEs reside at the embedding layer only; therefore, SpikeGPT still utilises some continuous activations. This implies that a LoRA module would still have to process continuous inputs rather than binary embeddings. LIF neurons can, however, process raw continuous inputs, which is more compatible and avoids the overhead of explicit encoding in non-spiking settings, but results in reduced biological plausibility.

## 2.3 Low-Rank Adaptation (LoRA)

Formally, $h$ as the output of the forward pass in a neural network with LoRA is defined as:

$$h = W_0 x + \Delta W_x, \tag{3}$$

where $W_0$ is the frozen weight matrix of the input activation vector $x$, and $\Delta W_x = BA$ is the low-rank weight matrix decomposition, where $B \in \mathbb{R}^{d \times r}$ and $A \in \mathbb{R}^{r \times k}$. The rank $r$ must be less

than $d$ and $k$ to ensure LoRA remains low-rank and efficient. The forward pass of a network is then calculated as usual, but during backpropagation, only the low-rank matrices are updated. LoRA typically applies a scaling factor $\alpha$ to control the magnitude of the low-rank update, and $\Delta W_x$ then becomes $\Delta W_x = \frac{\alpha}{r} BA$.

When fine-tuning with LoRA, some of the new dimensions introduced by the low-rank decomposition, referred to as *intruder dimensions*, may dominate the weight update. This can cause generalisation problems across domains (Shuttleworth et al., 2024). Intruder dimensions capture misleading correlations rather than learning generalisable features, which causes overfitting. On the other hand, it is also likely that some features in $\Delta W$ might be duplicated from $W$, which can amplify important features (Hu et al., 2022). This highlights the need for a method that is able to suppress misleading intruder dimensions while retaining useful amplifications.

## 3 SpikeLoRA Method

We aim to leverage the promising capabilities of SNNs and LoRA to develop a more robust parameter-efficient fine-tuning variant that is both energy-efficient on neuromorphic hardware and is less prone to overfitting. To this end, we introduce SpikeLoRA, based on the original LoRA definition (Hu et al., 2022), where $\Delta W$ (Eq.(3)) is modified as:

$$\Delta W = B \cdot (\mathcal{SN}(A) \odot A), \tag{4}$$

where $\mathcal{SN}$ is the LIF neuron that takes $A \in \mathbb{R}^{r \times k}$ as input such that $\mathcal{SN}(A) = \mathrm{LIF}(A) \in \{0, 1\}^{r \times k}$. The LIF neuron outputs a binary mask, which is applied to the original down projection from $A$ via element-wise product. This helps preserve previously learned information while zeroing out activations corresponding to the LIF nodes which did not spike (see Fig. 1). The inputs to the LIF neurons in the SpikeLoRA module are derived from continuous activations rather than explicitly employing encoding schemes such as rate or temporal encoding Eshraghian et al. (2023). As such, SpikeLoRA is applicable to both traditional and spiking LLMs (Zhu et al., 2024).

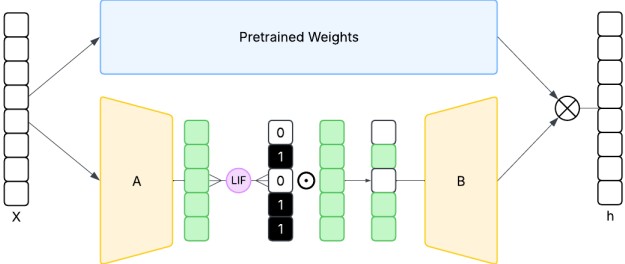

Figure 1: High-level diagram of SpikeLoRA during forward pass. For illustration purposes, only a vector is shown as input. In practice, a multi-dimensional tensor is passed as input. Adapted from the original LoRA definition (Hu et al., 2022).

In SpikeLoRA, each LoRA module is coupled with one LIF node, effectively gating the activations of the $A$-matrix (i.e., the *adapter-in* matrix). Since the LIF node resides in the low-rank space of the LoRA module, the element-wise multiplication has an $O(rk)$ complexity. If the $BA$-matrix (i.e. the *adapter-out* matrix) was gated such that $\Delta W = \mathcal{SN}(BA) \odot BA$, the complexity would increase to $O(dk)$, since $r \ll min(d, k)$ and $BA \in \mathbb{R}^{d \times k}$. Gating $BA$ would also bring $\alpha$, the scaling factor, into play, which can cause extreme sparsity or extreme magnitude updates. For this reason, we only gate the activations of the $A$-matrix.

Since our modification of LoRA is primarily based on the addition of a LIF neuron, SpikeLoRA can easily be coupled with other LoRA variations, such as AdaLoRA (Zhang et al., 2023). Similar to dropout, SpikeLoRA does not directly affect the underlying mechanisms of these variations, allowing for efficient cooperability. E.g., AdaLoRA's definition can be modified as:

$$\Delta = (\mathcal{SN}(P) \odot P) \cdot \Lambda \cdot Q, \tag{5}$$

where $\mathcal{SN}(P)$ is the spiking activation of the left singular vector $P$. Additionally, other matrices can be sparsified through spiking, similarly to how Lin et al. (2024) experimented with applying dropout to various parts of AdaLoRA. For a method like AdaLoRA, it is important to mention that the forward pass is mostly trivial, while the backwards pass requires careful handling to avoid double-counting gradient modifications. It is possible to detach the LIF neuron from the computational graph, but this impacts its ability to learn. We leave the exploration of adapting AdaLoRA and other LoRA methods with SpikeLoRA for future work.

Unlike stochastic activation suppression methods, SpikeLoRA is a trainable and adaptive method that directly targets the activations in the low-rank space. We hypothesise that SpikeLoRA's learned sparsity will amplify important features and suppress spurious and non-salient features. While directing the focus towards important features could potentially lead to fitting the problems better, we expect activation sparsification to implicitly aid in mitigating overfitting.

### 3.1 Why a LIF neuron?

While there are multiple methods to suppress activations, true learnable activation sparsity (i.e., true zeroes) is not always evident in these approaches. Approaches, such as a weighted sigmoid gate (Tanaka, 2020), target a single output in the activation tensor and do not take into account multiple activations (i.e., across batches), which the LIF neuron models in an event-driven way. The efficiency of the LIF neuron's temporal dynamics is evident as surveyed by Eshraghian et al. (2023), where event-driven learning allows for a reduction in activated neurons during learning in both forward and backward passes. Since the state of the LIF neuron is carried over multiple outputs, multiple activations can be zeroed as long as the membrane potential is less than the voltage threshold. This positions the LIF neuron as an effective mechanism to model and learn true learnable sparsity.

## 4 Experiments

To evaluate SpikeLoRA, we divide our experiments into three sections:

1. **SpikeLoRA on a traditional LLM (Section 4.1):** A traditional LLM is fine-tuned using the proposed SpikeLoRA module. We explore various setups to assess the impact of hyperparameters. We use the General Language Understanding Evaluation (GLUE) (Wang et al., 2019) benchmark to compare SpikeLoRA to classic LoRA.

2. **SpikeLoRA analysis (Section 4.2):** We discuss SpikeLoRA's characteristics, such as sparsity, efficiency, and its ability to mitigate overfitting.

3. **LoRA and SpikeLoRA on SpikeGPT (Section 4.3):** SpikeGPT is fine-tuned using both LoRA and SpikeLoRA. We utilise the NLU results from Zhu et al. (2024) to conduct a comparative performance analysis, and demonstrate the potential of an efficient spiking fine-tuning pipeline compatible with neuromorphic hardware.

For the traditional LLM, we use DeBERTaV3-Base (He et al., 2023) to conduct the experiments. For SpikeGPT, we make use of existing benchmark results (Zhu et al., 2024) as a baseline. LoRA and SpikeLoRA are applied to all linear layers in both DeBERTaV3-Base and SpikeGPT. Depending on availability, experiments were done using various Nvidia GPUs with at least 16GB VRAM. Each experiment is averaged over 5 independent runs with different seeds.

Unless otherwise stated, we use a learning rate warmup ratio of 6%, gradient clipping at 1.0, and a weight decay of 0.01. For LoRA, we use a rank of 8 and a dropout rate of 0. We found that the selection of the dropout rate did not significantly impact our findings. The results of different dropout rates are reported in Appendix A. We also use rsLoRA (Kalajdzievski, 2023) to stabilise the rank using the scaling factor $\alpha = \sqrt{r}$. By stabilising the rank with $\alpha$, rsLoRA enables a balanced tradeoff between fine-tuning efficiency and performance. The learning rate, batch size, and number of epochs are optimised per dataset, and reported in Appendix A.

For SpikeLoRA, we set the LIF's $V_\theta$ to 0.1, and report low-rank sparsity. We define sparsity as the percentage of zero activations after gating the $A$-matrix activations with the LIF neuron. Unless otherwise stated, sparsity values are reported as the average sparsity across all modules. Furthermore,

we use a soft reset function (Eshraghian et al., 2023) and set the leak constant, $\tau$, to 2.0 (Zhu et al., 2024).

The Corpus of Linguistic Acceptability (CoLA) dataset (Warstadt et al., 2019), which forms part of the GLUE benchmark, consists of only 8.5k training, 1043 validation, and 1063 test samples. Fine-tuning on CoLA, a small and skewed dataset, is susceptible to overfitting, and, in general, fine-tuning tends to perform worse on CoLA compared to other datasets in the GLUE benchmark (Zhang et al., 2023; Huang & Balestriero, 2024; Liu et al., 2024; Dettmers et al., 2023; Hu et al., 2022). We use the CoLA dataset in the majority of our experiments, as its small size and imbalanced label distribution provide a good test of robustness and generalisation when fine-tuning with SpikeLoRA. CoLA is evaluated using Matthew's correlation coefficient (Matthews, 1975), which is well-suited for imbalanced binary classification.

### 4.1 SPIKELORA ON A TRADITIONAL LLM

To establish the viability and competitiveness of SpikeLoRA, we fine-tune DeBERTaV3-Base with both LoRA and SpikeLoRA on the GLUE benchmark, and investigate the effects of varying voltage threshold (Section 4.1.1), rank (Section 4.1.2), and learning rate (Section 4.1.3) on the SpikeLoRA's performance.

#### 4.1.1 DIFFERENT $V_\theta$

Prior to performing comparisons with LoRA, we conduct an experiment to determine the appropriate $V_\theta$, i.e., voltage threshold value. Fig. 2 shows the effect of increasing $V_\theta$ for the CoLA dataset. Increasing $V_\theta$ causes an increase in the activation sparsity in the low-rank space. Since sparsity is desired, our goal is to maximise $V_\theta$ and the validation metric (e.g., accuracy), and minimise the validation loss. More formally, let $L_{V_\theta}$ be the validation loss, and $A_{V_\theta}$ be the validation metric using $V_\theta$:

$$A^* = \max(A_{V_\theta}), \qquad L^* = \min(L_{V_\theta}), \tag{6}$$

then the goal is to solve $\max(V_\theta)$ with the following constraints:

$$A_{V_\theta} \geq A^* - \delta_A, \qquad L_{V_\theta} \geq L^* + \delta_L, \tag{7}$$

where $\delta_A$ and $\delta_L$ are tolerated accuracy/loss parameters. In our experiments, we observed model collapse when $V_\theta \gtrsim 1.0$ (Fig. 2). When $V_\theta \leq 1.0$, minimal accuracy tradeoffs are made for increased sparsity (up to 97.24% during evaluation). As such, we conservatively set $V_\theta$ to 0.1 for the rest of the experiments. In Sections 4.1.2 and 4.1.3, with $V_\theta = 0.1$, we found that training starts with a global sparsity (average over each module per block) of $0.71 \pm 0.04$, and diverges from there on, depending on the setup.

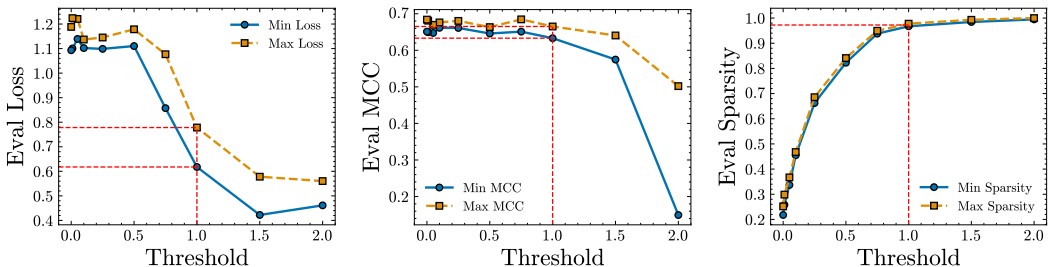

Figure 2: The effect of different $V_\theta$ threshold values on SpikeLoRA when fine-tuning on the CoLA dataset. Left: minimum and maximum evaluation loss across different $V_\theta$. Middle: Matthew's correlation coefficient (MCC) across different $V_\theta$. Right: global sparsity across all blocks as a result of the chosen $V_\theta$. The red lines indicate the minimum and maximum respective metrics when $V_\theta = 1.0$. Actual values and standard deviations are reported in Appendix A.

#### 4.1.2 DIFFERENT RANKS

Table 1 summarises the results of the experiments conducted to assess the effects of different ranks. We compare LoRA and SpikeLoRA to see if similar accuracy could be maintained by SpikeLoRA,

while introducing sparsity. Table 1 shows that SpikeLoRA performs better in most cases, except for ranks 1 and 4. For higher ranks, our results show that SpikeLoRA can better control overfitting (quantified via the generalisation gap) by introducing more sparsity, whereas LoRA's performance deteriorates beyond $r = 16$. This aligns with Mao et al. (2024): a higher rank can attain a richer representation, but is more susceptible to overfitting.

| Variant | Metric | $r = 1$ | $r = 2$ | $r = 4$ | $r = 8$ | $r = 16$ | $r = 32$ | $r = 64$ |
|---------|--------|---------|---------|---------|---------|----------|----------|----------|
| LoRA | $\text{CoLA}_{\text{MCC}}$ | **67.68** | 67.01 | **68.40** | 68.07 | 67.30 | 68.75 | 57.93 |
| | Gen. Gap | 0.75 | 0.83 | 0.85 | 0.80 | 1.13 | 1.14 | 1.44 |
| SpikeLoRA | $\text{CoLA}_{\text{MCC}}$ | $67.37_{-0.31}$ | $\mathbf{67.22}_{+0.12}$ | $67.47_{-0.93}$ | $\mathbf{68.37}_{+0.30}$ | $\mathbf{68.65}_{+1.35}$ | $\mathbf{69.43}_{+0.68}$ | $\mathbf{67.71}_{+9.78}$ |
| | Gen. Gap | $0.71_{-0.04}$ | $0.81_{-0.02}$ | $0.93_{+0.08}$ | $0.77_{-0.03}$ | $1.04_{-0.09}$ | $1.08_{-0.06}$ | $1.14_{-0.30}$ |
| | $\text{Sparsity}_{\%}$ | 33.47 | 43.67 | 52.75 | 69.74 | 73.83 | 80.87 | 92.69 |

Table 1: Matthew's correlation coefficient (MCC) and sparsity across blocks (%) when fine-tuning on the CoLA dataset using different ranks for LoRA and SpikeLoRA. The generalisation gap indicates the difference between evaluation and training loss (lower is better). Subscripts indicate SpikeLoRA's performance relative to LoRA, where green corresponds to improvement and red to reduction in performance. For each rank, the best accuracy is shown in bold.

### 4.1.3 DIFFERENT LEARNING RATES

Table 2 summarises the performance comparison between LoRA and SpikeLoRA on the CoLA dataset for various learning rates. It is evident from Table 2 that the sparsity of the low-rank space is directly proportional to the selected learning rate. When the learning rate is set between $1e-4$ and $5e-4$, the generalisation gap difference remains minimal. When the learning rate is set to $7e-4$ or higher, the generalisation gap of SpikeLoRA remains stable, while for LoRA it increases. Further, SpikeLoRA outperforms LoRA in terms of accuracy for all learning rate settings.

| Variant | Metric | 1e-4 | 3e-4 | 5e-4 | 7e-4 | 9e-4 |
|---------|--------|------|------|------|------|------|
| LoRA | $\text{CoLA}_{\text{MCC}}$ | 66.75 | 68.07 | 67.94 | 67.02 | 65.78 |
| | Gen. Gap | 0.66 | 0.80 | 1.08 | 1.15 | 1.17 |
| SpikeLoRA | $\text{CoLA}_{\text{MCC}}$ | $\mathbf{68.16}_{+1.41}$ | $\mathbf{68.37}_{+0.30}$ | $\mathbf{68.04}_{+0.08}$ | $\mathbf{67.39}_{+0.37}$ | $\mathbf{67.46}_{+1.68}$ |
| | Gen. Gap | $0.65_{-0.01}$ | $0.77_{-0.03}$ | $1.06_{-0.02}$ | $1.02_{-0.13}$ | $1.04_{-0.13}$ |
| | $\text{Sparsity}_{\%}$ | 51.56 | 69.74 | 77.88 | 82.58 | 85.49 |

Table 2: Matthew's correlation coefficient (MCC) and sparsity across blocks (%) when fine-tuning on the CoLA dataset using different learning rates for LoRA and SpikeLoRA. The generalisation gap indicates the difference between evaluation and training loss (lower is better). Subscripts indicate SpikeLoRA's performance relative to LoRA, where green corresponds to improvement and red to a reduction in performance. For each learning rate, the best accuracy is shown in bold.

### 4.1.4 GLUE BENCHMARK

Table 3 presents the results for GLUE when fine-tuning with LoRA and SpikeLoRA. It is evident from Table 3 that SpikeLoRA performed competitively, marginally outperforming LoRA for CoLA, SST-2, MRPC, and RTE, and performing comparably to LoRA for STS-B, MNLI, QNLI, and QQP. Table 3 also lists the sparsity achieved by SpikeLoRA per dataset, and shows that the resulting low-rank activations were at least $67\%$ sparse. We conclude that SpikeLoRA provides sparsification without a noticeable drop in performance metrics. Notably, our results suggest that SpikeLoRA performs better on smaller datasets (CoLA, MRPC, and RTE), indicating its potential to enhance generalisation in low-resource environments.

### 4.2 SPIKELORA ANALYSIS

In this section, we examine SpikeLoRA's behaviour to further understand its internal dynamics. We analyse SpikeLoRA in terms of sparsity (Section 4.2.1), regularisation (Section 4.2.2), and efficiency trade-offs (Section 4.2.3).

| Setup | CoLA$_{MCC}$ | SST-2$_{Acc}$ | MRPC$_{Acc/F1}$ | STS-B$_{Corr}$ | MNLI$_{Acc}$ | QNLI$_{Acc}$ | RTE$_{Acc}$ | QQP$_{Acc}$ | Avg. |
|---|---|---|---|---|---|---|---|---|---|
| LoRA | 68.07 | 95.55 | 89.41/92.47 | 91.36 | **90.44** | **94.17** | 86.07 | **91.83** | **88.36** |
| SpikeLoRA | **68.37** | 95.73 | 89.56/92.55 | 91.13 | 90.21 | 93.91 | 86.28 | 91.57 | 88.35 |
| Sparsity$_{\%}$ | 69.74 | 83.39 | 84.85 | 67.26 | 77.91 | 77.13 | 89.97 | 76.23 | 78.31 |

Table 3: GLUE benchmark comparison for fine-tuning using LoRA and SpikeLoRA. For each dataset, the best accuracy is shown in bold. The third row shows the sparsity of SpikeLoRA in the low-rank space. Average score across datasets is included.

### 4.2.1 SPARSITY

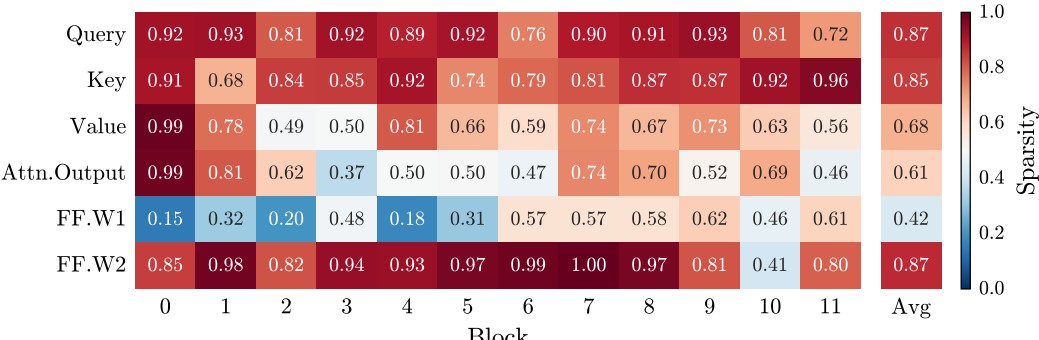

Figure 3: Sparsity of SpikeLoRA modules in various modules ($x$-axis) and blocks ($y$-axis) for the CoLA dataset.

Figure 3 presents a sparsity heatmap of LoRA trained on CoLA, organised in terms of individual modules and blocks. A traditional Transformer block consists of a query (Q), key (K), value (V), and output module within the self-attention head, as well as an FFN consisting of an intermediate layer (FF.W1) and an output layer (FF.W2) (Vaswani et al., 2017). We observe notably low sparsity in the intermediate layer compared to the other modules. Skean et al. (2025) noted that the intermediate layer signals encode richer information compared to other modules. As such, we conclude that learned sparsification is most beneficial in modules where richer information can be found. This conclusion is in line with our hypothesis that learned sparsity amplifies important features, i.e., features reach in information.

We tested a weighted sigmoid gate (WiG) (Tanaka, 2020) on LoRA and found that LoRA+WiG consistently outperformed both LoRA and SpikeLoRA (see Appendix A). The relative sparsity[1] induced by WiG, however, was not comparable to SpikeLoRA's sparsity. This led us to the conclusion that general activation suppression techniques are indeed a contributing factor to improving generalisation.

### 4.2.2 REGULARISATION

Results in Tables 1 and 2 demonstrate that SpikeLoRA in general tends to have a lower generalisation gap than LoRA. To further study the training dynamics, we investigated gradient norms for both methods. We found that SpikeLoRA's gradient norm maintains a moderately strong correlation with LoRA's (Pearson's $r = 0.678$, $p = 2.49e$-8), but with a slightly lower mean and standard deviation ($1.592 \pm 1.418$ (LoRA) down to $1.554 \pm 1.292$ (SpikeLoRA)). This shows that SpikeLoRA retains learning ability similar to LoRA while reducing extreme gradient updates, indicating that SpikeLoRA indirectly mitigates overfitting. We found this correlation to be particularly prevalent in higher-rank spaces (Table 1).

---

[1]We measured sparsity for WiG by counting activations smaller than 0.1 as "zeroed", which is a generous threshold.

We discovered that even when stochastic dropout is applied, SpikeLoRA consistently exhibits a smaller generalisation gap than LoRA while maintaining performance. This suggests that SpikeLoRA surpasses the effect of stochastic dropout, reducing overfitting without impacting performance. Results are reported in Appendix A.

### 4.2.3 EFFICIENCY

Despite promising results, we experienced a slight increase in training time when fine-tuning with SpikeLoRA: from $19.42 \pm 0.02$ minutes (LoRA) to $21.68 \pm 0.12$ minutes (SpikeLoRA) on average for the CoLA dataset. There are a couple of reasons why this is the case:

- LIF neurons introduce additional trainable parameters, although small in size (one neuron per module). Effectively, in large models, #Params$_{\text{SpikeLoRA}}$ $\approx$ #Params$_{\text{LoRA}}$. Kim et al. (2023) propose sharing LIF neurons across modules, which might reduce the number of LIF neurons and improve efficiency in future implementations.

- The additional $O(rk)$ complexity of scaling the output of the LIF with the original learned information contributes to the overall performance overhead of SpikeLoRA. The complexity, coupled with the LIF neuron, however, is not quadratic in nature.

- In our implementation, we used dense PyTorch layers, which regard zeroes as part of the computation. This means that the FLOPs are not reduced when scaling with the previously learned information.

To reduce the training time, quantisation techniques such as QLoRA (Dettmers et al., 2023) can be incorporated, which SpikeLoRA is compatible with. We briefly experimented with quantisation in Section 4.2.4.

As noted above, traditional hardware and dense layer implementation cannot fully exploit the benefits of sparsity. However, significant energy savings may be achievable should the model be deployed on neuromorphic hardware, where the process would benefit from sparsity.

### 4.2.4 SCALING SPIKELORA

To show the scaling of SpikeLoRA on larger models ($>$7B), we fine-tuned the QV weights of a 4-bit quantised version of Llama2-7B (Touvron et al., 2023) on the CoLA dataset. Because of time and resource constraints, we conducted a single run of 10 epochs on both LoRA and SpikeLoRA. Other than the number of epochs, we used the hyperparameters as reported in Appendix A.

After 10 epochs, LoRA's Matthews correlation coefficient (MCC) was 68.159% while SpikeLoRA's MCC was 66.292%, indicating a slight drop in performance. The sparsity of SpikeLoRA during evaluation was 74.157%, which is similar to our reported results on DeBERTa-v3 Base. We found that SpikeLoRA evaluated 17.206 samples per second, while LoRA only evaluated 16.852 samples per second. More so, on an Nvidia A40 with 48GB VRAM, LoRA used an average of 290.255 watts while SpikeLoRA used an average of 282.554 watts. These results suggest that coupling SpikeLoRA with a quantised model is the most efficient setup compared to LoRA.

Furthermore, we profiled the inference of SpikeLoRA (unmerged) and LoRA (merged) on the fine-tuned and quantised Llama2-7B. On CoLA's 1043 validation examples, SpikeLoRA exhibited an average latency of 13.706 ms (±0.195) per sample while LoRA exhibited an average latency of 11.866 ms (±0.241) per sample. In terms of computational cost, the profiler yielded $1.94 \times 10^{11}$ FLOPs per example (0.07% more FLOPs on SpikeLoRA).

### 4.3 FINE-TUNING SPIKEGPT WITH LORA AND SPIKELORA

We show the potential of fine-tuning SpikeGPT, which serves as the main inspiration for SpikeLoRA, with both LoRA and SpikeLoRA. Because of time and resource constraints, we limit our experiments to the subjectivity dataset (Pang & Lee, 2004).

Table 4 shows a performance comparison of different fine-tuning methods on SpikeGPT. As expected, full fine-tuning achieves the highest accuracy (95.30%), but is time-consuming, and tasks cannot be switched easily. LoRA and SpikeLoRA offer parameter-efficient alternatives to full fine-

tuning, and SpikeLoRA outperforms LoRA (+0.4%), indicating its potential effectiveness in a fully spiking fine-tuning pipeline. Notably, such a pipeline is compatible with neuromorphic hardware.

|  | SpikeGPT*† | Full FT* | LoRA | SpikeLoRA |
|---|---|---|---|---|
| Subj. | 89.10 | 95.30 | 90.70 | 91.10 |

Table 4: Fine-tuning performance for SpikeGPT on the subjectivity dataset (Pang & Lee, 2004). Results are measured using classification accuracy. * indicates numbers published by Zhu et al. (2024). † indicates that the SpikeGPT variants are trained from scratch on the respective dataset.

## 5 CONCLUSION & FUTURE WORK

Our work proposes SpikeLoRA, a sparse and efficient method to fine-tune large language models. We have shown that by adapting LoRA with a LIF neuron, it is possible to efficiently learn activation sparsity in the low-rank space. Through the empirical results, we have demonstrated that a high degree of sparsity (over $70\%$) can be achieved across blocks during fine-tuning, while maintaining performance comparable to or surpassing LoRA. Furthermore, as a side effect, we found that SpikeLoRA can mitigate overfitting, particularly in higher ranks and for smaller datasets, where LoRA is most susceptible to overfitting. These results suggest that spiking-inspired methods offer practical tools for efficient and robust low-rank adaptation.

By proposing SpikeLoRA, we not only contribute to the realm of parameter-efficient fine-tuning but also show that SNNs are ready to be integrated with current mainstream approaches. Assuming that SNNs and neuromorphic hardware become more widely adopted, SpikeLoRA may serve as the foundation for the next generation of efficient fine-tuning.

Future work includes investigating the effect of sparsity dynamics and designing an adaptive SpikeLoRA that controls sparsity by dynamically adjusting $V_\theta$ during training, similar to a learning rate scheduler. Similarly, making $V_\theta$ a learnable hyperparameter may also yield performance improvements. Furthermore, applying SpikeLoRA to embeddings, convolutional networks, graph neural networks, and multimodal LLMs presents exciting avenues for exploration. Future work also includes coupling SpikeLoRA with other methods, such as AdaLoRA and QLoRA, to further enhance efficiency. Finally, testing the proposed spiking pipeline on neuromorphic hardware is necessary to fully validate the approach.

### REPRODUCIBILITY STATEMENT

We are committed to ensuring that the results are reproducible in this paper. An implementation, along with scripts to run the benchmarks, will be released upon acceptance. Section 4 provides the training setup, while Appendix A provides the dataset-specific hyperparameters. These references provide sufficient information to replicate the results in this paper.

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

# A    ADDITIONAL DETAILS

For all GLUE experiments, we set the LoRA rank to 8, dropout to 0, and $V_\theta$ to 0.1. Table 1 shows the empirically selected hyperparameters used to perform GLUE experiments. For our $V_\theta$ and rank scaling analysis, we set the learning rate to $5e\text{-}4$.

| Task | Learning Rate | Batch Size | Epochs |
|------|---------------|------------|--------|
| CoLA | 3e-4 | 32 | 20 |
| SST-2 | 8e-4 | 64 | 4 |
| MRPC | 1e-3 | 32 | 20 |
| STS-B | 3e-4 | 16 | 8 |
| MNLI | 3e-4 | 64 | 3 |
| QNLI | 3e-4 | 32 | 3 |
| RTE | 1.2e-3 | 32 | 15 |
| QQP | 3e-4 | 64 | 3 |

Table 1: Hyperparameters for DeBERTA-v3 Base on the GLUE benchmark.

For $V_\theta$ scaling analysis on LoRA and SpikeLoRA ($r = 8$), we report the results in Table 2 (mean $\pm$ standard deviation). Relative drop in MCC is computed against the LoRA baseline.

| $V_\theta$ | Sparsity | MCC | Loss | Relative Drop |
|------------|----------|-----|------|---------------|
| 0.0 | $0.238 \pm 0.013$ | $0.666 \pm 0.013$ | $1.156 \pm 0.037$ | 0.018 |
| 0.01 | $0.279 \pm 0.017$ | $0.665 \pm 0.015$ | $1.170 \pm 0.049$ | 0.019 |
| 0.05 | $0.352 \pm 0.011$ | $0.657 \pm 0.010$ | $1.183 \pm 0.030$ | 0.031 |
| 0.1 | $0.461 \pm 0.005$ | $0.668 \pm 0.006$ | $1.124 \pm 0.014$ | 0.015 |
| 0.25 | $0.671 \pm 0.010$ | $0.666 \pm 0.008$ | $1.129 \pm 0.020$ | 0.018 |
| 0.5 | $0.835 \pm 0.007$ | $0.653 \pm 0.006$ | $1.150 \pm 0.029$ | 0.037 |
| 0.75 | $0.943 \pm 0.005$ | $0.663 \pm 0.013$ | $0.942 \pm 0.093$ | 0.022 |
| 1.0 | $0.972 \pm 0.005$ | $0.652 \pm 0.013$ | $0.720 \pm 0.070$ | 0.038 |
| 1.5 | $0.989 \pm 0.004$ | $0.614 \pm 0.032$ | $\mathbf{0.484 \pm 0.061}$ | 0.094 |
| 2.0 | $0.997 \pm 0.003$ | $0.351 \pm 0.168$ | $0.502 \pm 0.051$ | 0.482 |
| LoRA | - | $\mathbf{0.678 \pm 0.009}$ | $1.096 \pm 0.029$ | - |

Table 2: Evaluation metrics grouped by different $V_\theta$.

Figure 1 presents the investigation of the training dynamics of SpikeLoRA. Pearson correlation coefficient is 0.678 (p-value: $2.488e{-}8$), indicating a moderately strong correlation between the gradient norms of SpikeLoRA and LoRA.

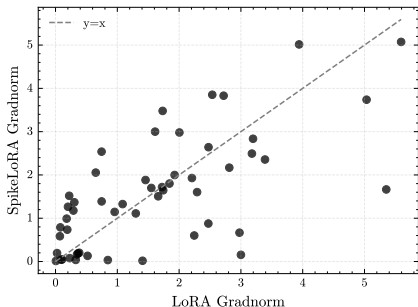

Figure 1: Scatterplot to show the gradnorm relationship between SpikeLoRA and LoRA.

Figure 2 and Table 3 show that SpikeLoRA consistently achieves a lower generalisation gap, whether or not dropout is applied, while maintaining similar performance.

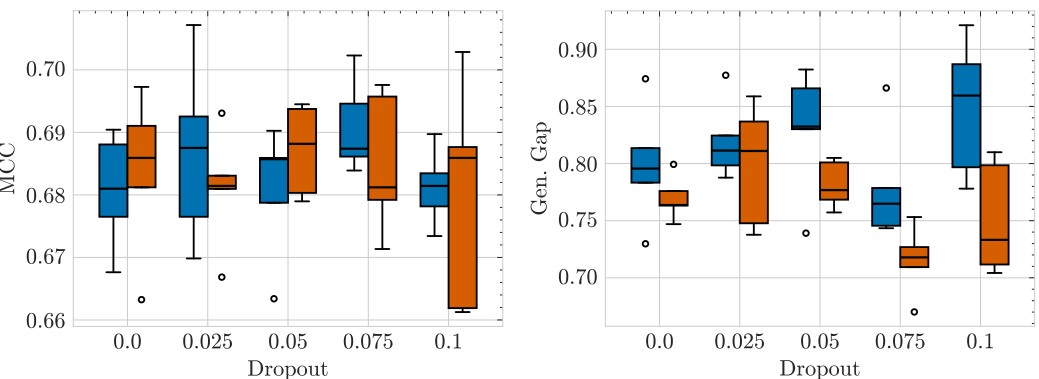

Figure 2: Boxplots showing the effect of dropout on MCC (higher is better) and generalisation gap (lower is better) for both LoRA (blue) and SpikeLoRA (brown).

| Dropout | Method | MCC | Gen. Gap |
|---------|--------|-----|----------|
| 0.0 | LoRA | $0.6807 \pm 0.0082$ | $0.7993 \pm 0.0467$ |
| | SpikeLoRA | $\mathbf{0.6837 \pm 0.0116}$ | $\mathbf{0.7698 \pm 0.0174}$ |
| 0.025 | LoRA | $\mathbf{0.6867 \pm 0.0130}$ | $0.8199 \pm 0.0313$ |
| | SpikeLoRA | $0.6811 \pm 0.0084$ | $\mathbf{0.7984 \pm 0.0480}$ |
| 0.05 | LoRA | $0.6808 \pm 0.0095$ | $0.8299 \pm 0.0496$ |
| | SpikeLoRA | $\mathbf{0.6871 \pm 0.0065}$ | $\mathbf{0.7817 \pm 0.0185}$ |
| 0.075 | LoRA | $\mathbf{0.6909 \pm 0.0067}$ | $0.7797 \pm 0.0451$ |
| | SpikeLoRA | $0.6850 \pm 0.0101$ | $\mathbf{0.7155 \pm 0.0270}$ |
| 0.1 | LoRA | $\mathbf{0.6812 \pm 0.0054}$ | $0.8485 \pm 0.0539$ |
| | SpikeLoRA | $0.6799 \pm 0.0161$ | $\mathbf{0.7515 \pm 0.0442}$ |
| $> 0.0$ | LoRA | $\mathbf{0.6849 \pm 0.0100}$ | $0.8195 \pm 0.0522$ |
| | SpikeLoRA | $0.6833 \pm 0.0113$ | $\mathbf{0.7618 \pm 0.0483}$ |

Table 3: Summary of different dropout rates applied to LoRA and SpikeLoRA when fine-tuning on the CoLA dataset.

Table 4 shows a partial GLUE benchmark comparison between SpikeLoRA, LoRA, and LoRA+WiG. The weighted sigmoid gate (WiG), proposed by Tanaka (2020), outperforms both methods while attaining a low relative sparsity percentage.

| Setup | CoLA$_{MCC}$ | SST-2$_{Acc}$ | MRPC$_{Acc/F1}$ | STS-B$_{Corr}$ | QNLI$_{Acc}$ | RTE$_{Acc}$ |
|-------|-----------|------------|--------------|-------------|-----------|----------|
| LoRA | 68.07 | 95.55 | 89.41/92.47 | 91.36 | 94.17 | 86.07 |
| LoRA+WiG | 67.72 | **95.85** | **89.80/92.69** | **91.40** | **94.22** | **86.79** |
| Rel. Sparsity$_\%$ | 1.21 | 11.98 | 9.58 | 4.41 | 2.48 | 10.86 |
| SpikeLoRA | **68.37** | 95.73 | 89.56/92.55 | 91.13 | 93.91 | 86.28 |
| Sparsity$_\%$ | 69.74 | 83.39 | 84.85 | 67.26 | 77.13 | 89.97 |

Table 4: Partial GLUE benchmark comparison for fine-tuning using LoRA, LoRA+WiG, and SpikeLoRA. For each dataset, the best accuracy is shown in bold. LoRA+WiG reports relative sparsity (%), while SpikeLoRA reports true sparsity (%). Relative sparsity were measured by counting activations smaller than 0.1 as "zeroed", which is a generous threshold.

