# OpenReview forum: "SpikeLoRA: Learnable Activation Sparsity for Low-Rank Adaptation using Spiking Neural Networks"
_ICLR.cc/2026/Conference — Submitted to ICLR 2026_

### Official Review · Reviewer_qCYf · 2025-10-31

**Soundness:** 3
**Presentation:** 3
**Contribution:** 2
**Rating:** 2
**Confidence:** 4

**Summary:**

The paper proposes SpikeLoRA, a new fine-tuning technique that combines Low-Rank Adaptation (LoRA) with spiking neural network (SNN) dynamics to introduce learnable activation sparsity. Instead of updating only low-rank weight matrices, SpikeLoRA integrates a Leaky Integrate-and-Fire (LIF) neuron that gates activations in the LoRA adapter, allowing the model to learn when to activate or suppress updates. This design aims to improve efficiency, reduce overfitting, and make LoRA-compatible models more suitable for neuromorphic hardware. The authors evaluate SpikeLoRA mainly on DeBERTaV3-Base using the GLUE benchmark—especially CoLA—and report over 70–90% activation sparsity with comparable or slightly improved performance relative to LoRA. The paper also presents limited results on SpikeGPT, showing that the method can operate in both traditional and spiking model pipelines.

**Strengths:**

The paper presents an original and well-executed idea by integrating Low-Rank Adaptation (LoRA) with spiking neural dynamics through a Leaky Integrate-and-Fire (LIF) neuron, creating a learnable activation sparsity mechanism. This cross-disciplinary combination is novel, technically sound, and clearly explained, with clean mathematical formulation and solid motivation. The experiments, though limited, show that SpikeLoRA can achieve high activation sparsity without significant loss in accuracy, suggesting potential regularization benefits. The paper is clearly written, easy to follow, and introduces a new perspective on efficiency—shifting from parameter-level to activation-level sparsity—which could have long-term significance for energy-efficient and neuromorphic fine-tuning research.

**Weaknesses:**

The paper’s main weakness is the limited experimental validation. Almost all results come from the CoLA dataset (Table 1–3), which is small and unbalanced. This makes it hard to judge if SpikeLoRA would generalize to larger or more diverse tasks. The full GLUE results in Table 3 are single runs, with no hyperparameter sweeps or statistical tests. There is no evidence that the gains hold beyond CoLA.

The authors show a large improvement at r = 64 in Table 1, but they do not explore this effect further. High-rank settings are rarely used in practice; the result looks interesting but anecdotal. The claim that SpikeLoRA “mitigates overfitting” (Section 4.2.2) is only supported by smaller generalization gaps. This is a weak proxy for real generalization. The paper does not show validation curves, variance across seeds, or ablations removing the LIF gate to prove causality.

Comparisons are also narrow. The only baseline is vanilla LoRA. Well-known LoRA extensions—AdaLoRA (Zhang 2023), DoRA (Liu 2024), QLoRA (Dettmers 2023), and LoRA Dropout (Lin 2024)—are not included, though several already address efficiency and regularization. Without these baselines, it is unclear if the benefit comes from the spiking mechanism or simply from added gating noise.

SpikeLoRA is tested on DeBERTaV3-Base and a small SpikeGPT task. Both are valid for proof-of-concept but too narrow for a method claiming broad applicability. There is no test on larger transformers (e.g., LLaMA, T5) or other modalities such as vision.

Finally, the claimed efficiency advantage is theoretical. Training time actually increases (Section 4.2.3), and no hardware or FLOP analysis is given. The paper would be stronger with runtime, energy, or sparsity-vs-accuracy trade-off plots.

Overall, the idea is novel but the evidence is thin. More models, baselines, and diagnostic experiments are needed to show that SpikeLoRA provides a consistent and meaningful improvement over existing LoRA variants.

**Questions:**

Most experiments rely on CoLA. Can you include results on larger or more diverse datasets to demonstrate that SpikeLoRA generalizes beyond a small benchmark? The performance differences on CoLA are marginal, which raises concerns about whether it is an appropriate dataset to support your claims.

In Table 1, SpikeLoRA shows a sharp performance gain at r = 64. Why does this occur, and is the effect consistent across datasets or random seeds?

Can you provide an ablation that removes or replaces the LIF gate to prove that the improvement comes from the spiking dynamics themselves rather than incidental gating noise?

Section 4.2.3 reports longer training time. Do you have runtime, FLOP, or hardware-level efficiency results to support the claim that SpikeLoRA is computationally efficient?

Have you evaluated SpikeLoRA on models beyond DeBERTaV3 and SpikeGPT, especially those that natively use LoRA (e.g., T5, LLaMA)? Results on such architectures would provide stronger evidence of generality.

---

> ### Author Response · Authors · 2025-12-03
>
> Thank you for taking the time to review our paper.
>
> > Most experiments rely on CoLA. Can you include results on larger or more diverse datasets to demonstrate that SpikeLoRA generalizes beyond a small benchmark? The performance differences on CoLA are marginal, which raises concerns about whether it is an appropriate dataset to support your claims.
>
> We show that SpikeLoRA generalises beyond a small benchmark by providing the complete GLUE benchmark results. Referring to our general response on testing WiG+LoRA, we show that LoRA+WiG consistently outperformed both LoRA and SpikeLoRA, but also by a small margin. This indicates that activation sparsity in the low-rank is not necessarily meant to improve performance by large margins (which parameter-optimised LoRA variants showcase), but rather to improve generalisation. The LIF neuron positions itself to be efficient on neuromorphic hardware, which motivates our choice.
>
> > In Table 1, SpikeLoRA shows a sharp performance gain at r = 64. Why does this occur, and is the effect consistent across datasets or random seeds?
>
> We hypothesise that SpikeLoRA effectively learns an optimal rank smaller than 64 through sparsity. We consider r=64 the point of failure for LoRA. Validation collapse at a high rank is a well-known problem in the vanilla LoRA (Liu et al., 2025, Wang et al., 2024), and served as motivation for multiple LoRA variants in the PEFT landscape. Table 1 (and all other results) reports the average over 5 independent seeded runs.
>
> > Can you provide an ablation that removes or replaces the LIF gate to prove that the improvement comes from the spiking dynamics themselves rather than incidental gating noise?
>
> Please see our general response for the motivation behind using spiking neurons. We show in Appendix A that adding stochastic gating noise by increasingly higher dropout rates is insufficient to surpass SpikeLoRA.
>
> > Section 4.2.3 reports longer training time. Do you have runtime, FLOP, or hardware-level efficiency results to support the claim that SpikeLoRA is computationally efficient?
>
> Please see our general response with regard to the conceptual energy-efficiency motivation.
>
> > Have you evaluated SpikeLoRA on models beyond DeBERTaV3 and SpikeGPT, especially those that natively use LoRA (e.g., T5, LLaMA)? Results on such architectures would provide stronger evidence of generality.
>
> Please see our general response regarding fine-tuning on Llama, specifically Llama2-7B.

---

### Official Review · Reviewer_CH7W · 2025-11-01

**Soundness:** 2
**Presentation:** 2
**Contribution:** 1
**Rating:** 2
**Confidence:** 5

**Summary:**

This paper proposes SpikeLoRA, a PEFT method that aims to improve LoRA by introducing activation sparsity through spiking neurons. The method inserts a LIF neuron between the $A$ and $B$ matrices of the LoRA decomposition. The binary output of this LIF neuron is used as a gating mask, applied via element-wise multiplication to the activations from matrix $A$, thereby achieving learnable sparsity.

The authors claim this method can achieve high activation sparsity (>70%) without significant performance degradation, serves as a regularization technique to mitigate overfitting, and offers compatibility with neuromorphic hardware.

**Strengths:**

1.  Reasonable Motivation for LoRA Regularization: The motivation to introduce learnable activation sparsity into PEFT as a form of regularization, specifically to combat overfitting in high-rank LoRA settings, is acceptable.
2.  Demonstrated Regularization Effect: The experiments (e.g., the generalization gap analysis in Table 3, Appendix A) provide some evidence that SpikeLoRA can offer a better regularization effect than standard LoRA, even when the latter uses Dropout.

**Weaknesses:**

1.  Misplaced Application and Limited Contribution: The paper's core contribution appears limited and ill-timed for both of its target communities:
    * For the SNN Community: Given that both Spiking LLMs and neuromorphic hardware are **far from mature**, introducing a PEFT technique like LoRA at this stage with SNN properties is premature and offers little immediate value.
    * For the ANN Community: When stripped of its "potential efficiency on future hardware" argument, the method degrades into a complex (using LIF) and inefficient (non-mergeable at inference) mechanism JUST for sparse regularization. This is largely unnecessary, as simpler, more efficient, and well-established ANN-native regularization techniques (e.g., Dropout, or other learnable ANN Heaviside-like functions) already exist or can be explored.
2.  Poorly Motivated Mechanism: Similar to the prior, the justification for using a LIF neuron as the gating mechanism is weak. Intuitively, I did not see any compelling reason why the neural dynamics of SNN and the LoRA task should be combined; Empirically, I also did not see evidence showing that the temporal dynamics of the LIF are playing a role. It functions merely as a complex hard gate with a fixed threshold $V_\theta$ and a surrogate gradient, failing to justify why this "bio-inspired" mechanism is superior to a simpler, less computationally expensive **ANN learnable gate** (e.g., a thresholded ReLU or Sigmoid).
3.  Efficiency Shortages:
    * Training Overhead: The authors acknowledge (Sec 4.2.3) that SpikeLoRA is **slower** to train (by ~11%) than standard LoRA on GPUs, due to extra computations and the inability to leverage sparse operations in dense libraries.
    * Inference Overhead: This is the most critical flaw. Unlike standard LoRA (whose $\Delta W$ can be merged into $W_0$ to eliminate all inference overhead, SpikeLoRA's non-linear LIF gate makes merging impossible (especially LIF itself can only perform serial calculations). This means it *permanently* adds inference latency and computational cost.
4.  Insignificant Performance Gains: The experimental results do not justify the added complexity. On the GLUE benchmark (Table 3), the average performance of SpikeLoRA is **nearly identical** to vanilla LoRA.
5.  Unfair Baseline Comparison: As the main claimed benefit is regularization, the most direct baseline is Dropout. However, the LoRA baseline in the main tables uses `dropout=0`, meaning SpikeLoRA (with added complexity) was only compared to vanilla LoRA instead of other regularization methods, which is an unfair comparison.

**Questions:**

1.  Necessity of LIF and Effect of Neural Dynamics of SNN: Does SpikeLoRA actually utilize the temporal dynamics of the LIF neuron? If not, please justify why a simpler learnable gating function (e.g., a surrogate gradient for $g(x) = \text{ReLU}(x - \theta)$) was not used instead.
2.  Quantification of Inference Overhead: Please provide a concrete comparison of the inference overhead (in FLOPs and wall-clock time) between SpikeLoRA (which cannot be merged) and a standard LoRA model *after* its weights have been merged.
3.  Comparison with Optimal Dropout: Please provide a direct performance comparison on the GLUE benchmark between SpikeLoRA (e.g., `dropout=0`, $V_\theta=0.1$) and standard LoRA configured with its *optimal* dropout rate (e.g., 0.05 or 0.075 from Appendix A).
4.  $V_\theta$ Selection: Why is the threshold $V_\theta$ a fixed hyperparameter? Was making it a learnable parameter, or adapting it per-layer, ever attempted?

---

> ### Author Response · Authors · 2025-12-03
>
> Thank you for taking the time to review our paper.
>
> > Necessity of LIF and Effect of Neural Dynamics of SNN: Does SpikeLoRA actually utilize the temporal dynamics of the LIF neuron? If not, please justify why a simpler learnable gating function (e.g., a surrogate gradient for g(x) = ReLU(x - theta)) was not used instead.
>
> SpikeLoRA doesn’t utilise the temporal dynamics of LIF neurons, as the pipeline surrounding it is continuous. Our focus was to bring biologically plausible methods to the PEFT realm, and a LIF neuron model is perfectly suited to model true learnable sparsity. We suspect that attempting to enforce true stochastic sparsity by using a dropout mechanism will likely become unstable as the dropout rate increases.
>
> > Quantification of Inference Overhead: Please provide a concrete comparison of the inference overhead (in FLOPs and wall-clock time) between SpikeLoRA (which cannot be merged) and a standard LoRA model after its weights have been merged.
>
> We tested the inference of SpikeLoRA (unmerged) and LoRA (merged) on a quantised version of Llama2-7B when fine-tuned on the CoLA dataset (QV weights only). On 1043 validation examples, SpikeLoRA exhibited an average latency of 13.706 ms (±0.195) per sample, while LoRA exhibited an average latency of 11.866 ms (±0.241) per sample. We profiled inference for both SpikeLoRA and LoRA, where the profiler yielded 1.94x10^11 FLOPs per example (0.07% more FLOPs on SpikeLoRA).
>
> > Comparison with Optimal Dropout: Please provide a direct performance comparison on the GLUE benchmark between SpikeLoRA (e.g., dropout=0) and standard LoRA configured with its optimal dropout rate (e.g., 0.05 or 0.075 from Appendix A).
>
> The effects of SpikeLoRA and its LIF neuron in the low-rank space are isolated by using the same baseline hyperparameters. We primarily used the hyperparameters selected by Ding et al. (2023) and therefore did not perform exclusive hyperparameter fine-tuning. Furthermore, if we were to use an optimal dropout rate, we’d have to do the same to SpikeLoRA, as SpikeLoRA is also compatible with dropout. In Appendix A, we show that dropout doesn’t impact our findings with regard to regularisation.
>
> > V_t Selection: Why is the threshold, V_t, a fixed hyperparameter? Was making it a learnable parameter, or adapting it per-layer, ever attempted?
>
> We agree that fixing V_t may be limited in a similar way that AdaLoRA (Zhang et al., 2023) shows notable performance gains over baselines when using a dynamic rank instead of a fixed rank. We therefore conclude that while we briefly mentioned a scheduler-like approach, per-layer and learnable methods may also be worth exploring. This may yield both performance and efficiency gains.

---

### Official Review · Reviewer_N1ZU · 2025-11-01

**Soundness:** 3
**Presentation:** 3
**Contribution:** 3
**Rating:** 4
**Confidence:** 4

**Summary:**

This paper introduces SpikeLoRA, a spiking neural network-inspired extension to Low-Rank Adaptation (LoRA) for parameter-efficient fine-tuning of large language models. SpikeLoRA introduces the leaky integrate-and-fire (LIF) neuron as a learnable, biologically inspired activation gate within the low-rank space of LoRA modules, aiming to induce sparsity in activations while preserving information. The approach is empirically evaluated across standard language model benchmarks, with analyses of sparsity, efficiency, regularization, and overfitting mitigation, and includes comparative experiments on both traditional and spiking LLMs such as DeBERTaV3-Base and SpikeGPT.

**Strengths:**

s1:Integrating LIF neurons directly into the LoRA pipeline introduces a learnable mechanism for controlling activation sparsity. This is a novel perspective compared with existing parameter-level or dropout-based sparsification approaches

**Weaknesses:**

w1: The paper hypothesizes that SpikeLoRA amplifies salient features and suppresses noise but lacks a formal analysis connecting LIF-based sparsity to representational power or generalization. The operational regime of the LIF neuron and the risks of over- or under-sparsification are explored only empirically (see Section 3, Section 4.1.1, Figure 2). A probabilistic or theoretical characterization of this trade-off would strengthen the work.

w2: The paper does not test whether the LIF neuron itself is responsible for the improvements. Comparisons to simpler gating methods (e.g., hard thresholds or surrogate non-biological gates) are missing, leaving unclear whether the benefits stem from the spiking dynamics or from general sparsity enforcement.

**Questions:**

q1: Do you have evidence—analytical or empirical—of real energy or speed gains on neuromorphic or low-power hardware compared to LoRA and other sparsification methods?

q2: What happens if the LIF neuron is replaced with simpler gating mechanisms (e.g., hard-threshold or continuous surrogates)? Are the gains biologically unique or general to sparse gating?

---

> ### Author Response · Authors · 2025-12-03
>
> Thank you for taking the time to review our paper.
>
> >w1: The paper hypothesizes that SpikeLoRA amplifies salient features and suppresses noise but lacks a formal analysis connecting LIF-based sparsity to representational power or generalization. The operational regime of the LIF neuron and the risks of over- or under-sparsification are explored only empirically (see Section 3, Section 4.1.1, Figure 2). A probabilistic or theoretical characterization of this trade-off would strengthen the work.
>
> We believe that correlating LIF-based sparsity to representational power would purely be problem-specific. It’s evident from our GLUE benchmark results (Section 4.1.4) that sparsity does not always relate to performance. Sparsity is purely an efficiency characteristic of spiking neural networks (also commonly referred to as the “3 S’s of SNNs”, namely, spikes, sparsity, and static suppression (Eshraghian et al., 2023)).
>
> > w2: The paper does not test whether the LIF neuron itself is responsible for the improvements. Comparisons to simpler gating methods (e.g., hard thresholds or surrogate non-biological gates) are missing, leaving unclear whether the benefits stem from the spiking dynamics or from general sparsity enforcement.
>
> See our general response for the motivation behind using spiking neurons and a comparison to simpler gating methods.
>
> > q1: Do you have evidence—analytical or empirical—of real energy or speed gains on neuromorphic or low-power hardware compared to LoRA and other sparsification methods?
>
> We don't have empirical evidence of the real energy or speed gains compared to LoRA. Neuromorphic hardware is currently limited in its capacity to fine-tune large-scale LLMs, preventing an empirical evaluation. However, the FLOPs reduction may be more or less equivalent to the sparsity percentage. Analytically, however, it is demonstrated that SNNs are theoretically more energy-efficient and are actively being applied to resource-constrained real-world applications (Eshraghian et al., 2023).
>
> > q2: What happens if the LIF neuron is replaced with simpler gating mechanisms (e.g., hard-threshold or continuous surrogates)? Are the gains biologically unique or general to sparse gating?
>
> See our response on w2.

---

### Official Review · Reviewer_5XEU · 2025-11-03

**Soundness:** 2
**Presentation:** 3
**Contribution:** 2
**Rating:** 4
**Confidence:** 2

**Summary:**

The paper introduces a LoRA-based fine-tuning method that incorporates learnable spiking-neuron gates into the low-rank adaptation path. Instead of updating weights densely, SpikeLoRA uses a Leaky-Integrate-and-Fire gating mechanism to sparsify LoRA activations, enabling both parameter-efficient and activation-efficient tuning without modifying base weights. Experiments show that this spiking gate can learn task-relevant sparse patterns, achieving over 70% activation sparsity while maintaining performance, and sometimes even improving accuracy in low-data or high-rank settings by reducing overfitting. The method also extends naturally to spiking language models like SpikeGPT, demonstrating compatibility with neuromorphic compute and highlighting energy-efficiency potential.

**Strengths:**

The paper presents a new integration of spiking neuron gating into LoRA, creatively bringing neuromorphic concepts into mainstream LLM fine-tuning. The introduction of learnable LIF gates to induce adaptive activation sparsity in low-rank updates is both unique and meaningfully expands the landscape of parameter-efficient tuning. The architecture, gating dynamics, and training procedure are clearly described, supported by helpful diagrams comparing standard LoRA and SpikeLoRA. The motivation around energy efficiency and biological inspiration is articulated clearly and remains balanced without overstatement.

**Weaknesses:**

- The motivation for combining spiking neurons with LoRA is not fully convincing from a practical perspective. While biological inspiration and neuromorphic alignment are emphasized, the paper does not clearly articulate a strong need for spiking gating in mainstream LLM fine-tuning. For example, existing sparsity-inducing PEFT methods (e.g., sparse/structured LoRA variants, gating via ReLU/Hard-Concrete, learned token- or head-level sparsity) already enable activation reduction without introducing spiking dynamics. A clearer argument for why spiking-based gating is fundamentally preferable can be helpful.

- The energy-efficiency motivation remains largely conceptual. The experiments focus on sparsity and accuracy, but do not provide direct measurements of energy savings (e.g., FLOPs, inference energy estimates, wall-power measurements on standard hardware, or neuromorphic deployment benchmarks).

- The applicability to real neuromorphic systems is asserted but not fully demonstrated. Although SpikeGPT experiments show compatibility in principle, the paper lacks an end-to-end demonstration on neuromorphic hardware or a quantitative comparison to alternative energy-efficient architectures (e.g., binarized activations, event-driven RNNs).

**Questions:**

- Is the spiking gate applied uniformly across all LoRA modules optimal? Could selective placement (e.g., only in attention projections or specific layers) yield similar sparsity with lower complexity?


- Can the authors comment on the scalability of SpikeLoRA to larger models (e.g., >7B parameters)? Since neuromorphic motivations matter most at larger scales, empirical or theoretical discussion on scaling behavior would be valuable.

- How sensitive is the method to the LIF parameters (leak constant, threshold, reset function)? The paper notes learnability, but ablations or analysis on stability and convergence would help justify design choices.

---

> ### Author Response · Authors · 2025-12-03
>
> Thank you for taking the time to review our paper.
>
> > The motivation for combining spiking neurons with LoRA is not fully convincing from a practical perspective. While biological inspiration and neuromorphic alignment are emphasised, the paper does not clearly articulate a strong need for spiking gating in mainstream LLM fine-tuning.
>
> Please see our general response for the motivation behind using spiking neurons.
>
> > Conceptual energy-efficiency motivation
>
> Please see our general response with regard to the energy-efficiency motivation.
>
> > Is the spiking gate applied uniformly across all LoRA modules optimal? Could selective placement (e.g., only in attention projections or specific layers) yield similar sparsity with lower complexity?
>
> We agree that this is a valid point. We believe that focusing on certain layers where sparsity is low (e.g., as shown in Figure 3) may yield on-par performance with lower complexity. We believe that coupling SpikeLoRA with LoRA variants, such as AdaLoRA (Zhang et al., 2023), may yield even better results with lower computational costs. The paper is intended to serve as a proof of concept to provide a foundation for subsequent studies to delve deeper into variants, combinations, and hyperparameters.
>
> > Can the authors comment on the scalability of SpikeLoRA to larger models (e.g., >7B parameters)? Since neuromorphic motivations matter most at larger scales, empirical or theoretical discussion on scaling behaviour would be valuable.
>
> When employing a LIF neuron, no quadratic computations are performed. We considered the GLUE benchmark as an indication of how SpikeLoRA performs across different datasets. The LIF neuron simply adds proportional overhead (as per definition) and doesn’t scale quadratically as the model size increases. For more concrete evaluations, please see our general response with regard to larger models.
>
> > How sensitive is the method to the LIF parameters (leak constant, threshold, reset function)? The paper notes learnability, but ablations or analysis on stability and convergence would help justify design choices.
>
> We perform a univariate scaling analysis on different voltage thresholds in Section 4.1.1 to evaluate the sensitivity of different thresholds to the performance metrics and sparsity. We followed standard practice with the reset function by using a soft reset function (Eshraghian et al., 2023). The leak constant, 𝜏, was set to the same value used in SpikeGPT (Zhu et al, 2024).

---

### Author Response · Authors · 2025-12-03
**Addressing General Comments**

We’d like to thank all the reviewers for their comprehensive comments and suggestions. We worked through the feedback and updated the paper accordingly. Here follows a general response addressing some of the most highlighted concerns:

## Conceptual energy-efficiency motivation

The end-to-end pipeline in the spiking and non-spiking LLMs that we used to conduct our experiments (SpikeGPT and DeBERTa-v3 Base) is continuous (i.e. continuous → binary (LIF) → continuous). The actual effect of the LIF neuron will be difficult to isolate, and it will be similarly difficult to calculate the analytical speed-ups, as there’s no true spiking input (e.g., by means of rate or temporal encoding). Since we’re adding a LIF neuron to LoRA, which maintains internal states (such as membrane potential), accumulates the potential sequentially over batches, and forms part of the computational graph, it is sensible to expect additional computational overhead on the traditional von Neumann architecture. However, we believe that sparsity would allow us to balance the overhead with neuromorphic hardware. Scaling with A to preserve learned representations, for example, is a tedious computation that is simply done to rescale the down projection. In a fully spiking environment, this won’t be necessary, as encoding methods will be used.

Our motivation will remain purely conceptual/theoretical as neuromorphic hardware is currently limited in its capacity to fine-tune large-scale LLMs, preventing an empirical evaluation. However, we suspect that the FLOPS reduction may be proportional to the sparsity percentage.

## SNN motivation and a comparison of the LIF neuron to simpler gating methods

A couple of reviewers questioned the core motivation behind the LIF neuron and why simpler gating methods (such as a learnable sigmoid) were not chosen instead.

We agree that sparse (gating) techniques exist which do take care of activation reduction, but true activation sparsity (i.e., true zeroes) is not always evident in these approaches. These approaches target a single output in the activation tensor and do not take into account multiple activations (i.e., across batches), which the LIF neuron models in an event-driven way. The efficiency of the LIF neuron’s temporal dynamics is evident as surveyed by Eshraghian et al. (2023), where event-driven learning allows for a reduction in activated neurons during learning in both forward and backward passes. Since the state of the LIF neuron is carried over multiple outputs, multiple activations can be zeroed while the membrane potential is less than the voltage threshold.

In response to these comments, we tested a weighted sigmoid gate (WiG) (Tanaka, 2020) and found that LoRA+WiG consistently outperformed both LoRA and SpikeLoRA. The discussion and results of LoRA+WiG can be found in Section 3.1, Section 4.2.1, and Appendix A of the updated paper.

We acknowledge the fact that spiking gating in mainstream fine-tuning approaches might not be empirically groundbreaking in terms of efficiency and performance. However, we have a strong belief that SpikeLoRA may thrive in efficient edge AI fine-tuning end-to-end spiking pipelines, given that neuromorphic hardware is used. Bringing SNNs further into the ANN community (especially LLM and PEFT), although not yet fully mature and efficient, will further drive research on biologically plausible learning, which is already a critical requirement in low-power edge AI applications.

## SpikeLoRA on a 7B+ model

A couple of reviewers requested information regarding the scalability of SpikeLoRA by testing it on a model with more than 7 billion parameters. In response to these comments, we fine-tuned SpikeLoRA and LoRA on the QV weights of a 4-bit quantised version of Llama2-7B using the CoLA dataset. We report the results in Section 4.2.4 of the updated paper.

## Resource-constrained study

While we would most certainly like to perform a lot of experiments to further support our motivation and findings, we were short of computing resources both in time and capacity. Consequently, our paper is focused on a proof of concept combining SNNs with LoRA. In the main paper, we present three major univariate scaling analyses on different voltage thresholds, learning rates, and ranks. We also benchmarked SpikeLoRA on the GLUE datasets to properly validate our approach. All results (except for the SpikeGPT and Llama2-7B results) are reported as an average of 5 seeded runs.

Furthermore, we do not currently have access to neuromorphic hardware, and as such, the practical verification of neuromorphic speed-ups is left for future work.

---

### Meta-Review · Area_Chair_sqK7 · 2026-01-06

**Summary:**

This paper proposes SpikeLoRA, a novel extension of LoRA that introduces learnable activation sparsity via a LIF-based spiking gate in the low-rank space. Reviewers generally agree that the idea is original and technically sound, and appreciate the clear formulation and cross-disciplinary perspective. At the same time, reviewers raise concerns about the practical necessity of spiking neurons over simpler gating mechanisms, the largely conceptual nature of the energy-efficiency claims, and the limited empirical scope and baselines. Overall, the work is viewed as an interesting proof-of-concept, but with mixed opinions on its readiness and impact for the ICLR main program.

**Reviewer Concerns:**

The rebuttal addresses several key concerns by clarifying that the neuromorphic motivation is conceptual, adding experiments on a quantized Llama2-7B model. These additions improve transparency and positioning. However, some concerns remain outstanding, particularly regarding the lack of demonstrated practical efficiency gains, the limited advantage of LIF gating compared to simpler ANN-native alternatives, and the relatively narrow experimental validation and baselines.

**Reviewer Scores:**

The score distribution would likely remain mixed, centered around borderline reject to weak accept.

---

### Decision · Program_Chairs · 2026-01-26

Reject